# Costs of outpatient services at selected primary healthcare centers in Bangladesh: A cross-sectional study

Md. Zahid Hasan[1,2,3]*, Gazi Golam Mehdi[1], Khadija Islam Tisha[1], Md. Golam Rabbani[1], Mohammad Wahid Ahmed[1], Subrata Paul[4], Ziaul Islam[1], Shehrin Shaila Mahmood[1]

1 Health Economics and Financing, Health Systems and Population Studies Division, icddr,b, Dhaka, Bangladesh, 2 Nuffield Centre for International Health and Development, Leeds Institute of Health Sciences, University of Leeds, Leeds, United Kingdom, 3 Academic Unit of Health Economics, Leeds Institute of Health Sciences, University of Leeds, Leeds, United Kingdom, 4 Health Economics Unit, Health Services Division, Ministry of Health and Family Welfare, Dhaka, Bangladesh

* md.zahid@icddrb.org

## Abstract

### Background

Upazila Health Complexes (UzHC) serve as the backbone of primary healthcare (PHC) at the sub-district level in Bangladesh, delivering comprehensive healthcare services including both inpatient and outpatient services to the grassroots levels. However, not all the pre-scribed medicines and diagnostics services are always available at these facilities for outpatient care. This results in out-of-pocket expenditure (OOPE) to the patients for getting prescribed medicines and diagnostics services which has not been properly explored. Thus, we aimed to estimate the overall provider and user costs for outpatient care services at selected UzHCs in Bangladesh.

### Methods

An ingredient-based costing approach was applied to estimate the costs for the most commonly reported illnesses at outpatient of UzHCs from a societal perspective. We conducted a health facility survey at four purposively selected UzHCs to estimate provider costs and a patient exit survey among 452 patients of selected illnesses to estimate the user costs. Commonly reported illnesses were identified in consultation with healthcare providers of these facilities. The difference between costs of prescribed and provided medicines at UzHCs was estimated using the market prices. Data was collected between February to March 2021.

### Results

The societal costs of the common outpatient illness or symptoms varied significantly, ranging from BDT 642 to BDT 1,384 per episode. Antenatal care had the highest cost burden at BDT 1,384, followed by respiratory illness at BDT 783 and urinary tract infection at BDT 670. On average, the provider spent BDT 289 for treating an outpatient, while a patient

**Data Availability Statement:** The datasets used and/or analyzed during the current study are included as supplementary data file

**Funding:** The study was funded by the Health Economics Unit (HEU), Health Services Division, Ministry of Health and Family Welfare, the Government of Bangladesh (Grant #: GR-01964).

**Competing interests:** The authors have declared that no competing interests exist.

incurred BDT 446 as OOPE. Further, a patient was expected to spend an average of BDT 341 for purchasing medicines not provided from UzHCs.

## Conclusion

Our study found significant gaps between prescribed and provided medicines at UzHCs, leading to higher OOPE for patients. The current healthcare resource allocation strategy does not consider the outpatient load and healthcare demand at PHC facilities, which further exacerbates this gap. Addressing this gap requires a fundamental shift towards a demand-driven resource allocation model within the healthcare financing strategy to improve healthcare access and achieve health for all.

## Introduction

Primary healthcare (PHC) system serves as the cornerstone for delivering essential health services, promoting preventive care, and addressing health disparities at the community level [1]. However, in many low- and middle-income countries (LMICs) public financing for healthcare services is often insufficient, and risk-pooling mechanisms are limited. In these countries people face high out-of-pocket expenditure (OOPE) while accessing healthcare, especially from private sectors. Given the epidemiological transition, disease outbreaks, and inadequate governance structures in LMICs, a significant gap exists in the delivery of PHC services as per needs of the population [2]. Public healthcare services are subsidized and much lower expensive than those provided by private facilities. For such reason, people from lower socioeconomic groups prioritize public facilities as the first contact point for accessing PHC services. Improving access to quality healthcare services at affordable costs for all groups of population is the key theme for achieving universal health coverage (UHC). Thus, strengthening the public PHC system is vital for achieving UHC in LMICs.

Bangladesh has a multi-tiered healthcare delivery system in the public sector for its citizens i.e., primary, secondary, and tertiary/specialized care [3]. The government is the key actor in providing healthcare services in the pluralistic health systems of Bangladesh. However, the country allocates only 2.88% of its gross domestic product (GDP) to the health sector, with government expenditure relative to GDP at 0.66%. This places the country among the lowest health spenders in the Southeast Asian Region [4]. At the public PHC level, there are 424 Upazila Health Complexes (UzHCs), 87 Union Health and Family Welfare Centers, 1,312 Union Sub-Centers, and 13,948 Community Clinics [5]. UzHCs offer comprehensive PHC services, including outpatient, inpatient, emergency care, and serve as the referral center for lower level facilities (e.g., Union Health and Family Welfare Centers and Community Clinics), which solely focus on outpatient services [6]. In these facilities, the patients pay nominal fees for user/registration and diagnostic services, while essential medicines are typically provided free as per availability [7]. The National Health Policy 2011, prioritized essential drugs and basic medical services in all UzHCs, yet resources are not allocated as per the population needs [8]. Although UzHCs provide both inpatient and outpatient healthcare services, the current resource allocation mechanism, determined by the number of available inpatient beds in a health facility, fails to consider the outpatient load [9]. For most illnesses, patients receive medicine from UzHCs for a limited period, based on availability, even when prescribed for a longer duration. Moreover, due to inadequate diagnostic services, UzHCs often fails to provide the prescribed diagnostic tests. This gaps in medicine provision can lead to non-compliance, result in disease

progression, lower quality of life, and increased strain on healthcare resources through frequent hospital visits and admissions. In addition, the OOPE for healthcare in Bangladesh is about 68.5% which is the highest among its neighboring countries [10]. Such high OOPE is a major barrier in accessing healthcare services [11, 12] which is further exacerbated by the limited resources available at UzHCs. Over 90% of patients at public health facilities use outpatient services [13].However, there is a gap in empirical evidence on the cost of accessing these services at public PHC facilities in Bangladesh. Considering the greater demand for outpatient services, assessing the costs of treating common diseases and the financial burden on patients can help policymakers identify opportunities and strategies to address these issues and accelerate progress toward UHC [14].

Several studies have been conducted to estimate the outpatient costs of healthcare services in similar contexts. For example, a 2019 study in India estimated the average cost per outpatient visit at public providers to be INR 400 (USD 5.60) [15]. Similarly, a 2015 study in Cambodia found that outpatient consultation costs at health centers ranged from USD 2.33 to USD 4.89 [16]. Few studies estimated the cost of outpatient services in public hospitals in Bangladesh [17, 18]. A 2011 study in Bangladesh estimated the average direct cost of outpatient treatment at public hospitals to be BDT 279 (USD 3.72) [18]. Another study from the same year found that pregnant women using MOHFW facilities incurred an average OOPE of BDT 565 (USD 7.53) for outpatient care [17]. Notably, there is a dearth of studies focusing on the cost of outpatient service at UzHCs. Therefore, the current study aimed to estimate the overall provider and user costs for outpatient PHC services at selected UzHCs in Bangladesh.

## Methods

### Study design and setting

This was a cross-sectional study supplemented by an ingredient-based costing approach, conducted in 2021 in the Chandpur district of Chattogram division. This district has a total of seven UzHCs. The quantitative method included a health facility survey and review of relevant costs document to estimate provider costs and a patient exit survey to assess the costs incurred by patients for utilizing outpatient services.

### Sample size

Since the primary focus of this study was on estimating the cost of outpatient service, a single population mean formula was used to determine the sample size. A prior study reported that the standard deviation of cost for outpatient service at public tertiary hospitals in Bangladesh was USD 124. We used this value to calculate the sample size for the UzHCs. Assuming a 95% confidence level and a 10% margin of error from the mean costs, we determined that 340 respondents would be required for the exit survey. Further, taking into account a 1.2 design effect and a 10% non-response rate, the estimated sample size increased to 512. This sample was proportionally allocated across the selected UzHCs based on the previous year's utilization rates. However, we could interview a total of 452 respondents during the time frame for data collection.

### Sampling technique

We collected cost data through surveys from both service providers and outpatient healthcare users.

**Selection of health facility for survey.**   We purposively selected Chandpur district for this study, which has a total of seven UzHCs. These UzHCs were categorized into two groups

based on the number of beds: 31-bedded (four) and 50-bedded (three) UzHCs. From each group, we selected two UzHCs based on their outpatient utilization rates—one with high utilization and one with low utilization. The outpatient utilization data for these facilities were extracted from the local health bulletin for the past three years (2018–2020) [19]. The selected 31-bedded UzHCs were Faridganj (low) and Matlab North (high), while the selected 50-bedded UzHCs were Hajiganj (high) and Kachua (low).

**Patient selection for exit survey.** To select patients for the exit survey, we initially prepared a list of the most frequent outpatient illness or symptoms. The list was prepared in consultation with the Upazila Health and Family Planning Officers and Medical Officers of the selected UzHCs. During the listing, different facilities reported different illness as most frequent. However, from the collected list, we selected the illness those were reported as common in majority of the UzHCs to keep consistent with the type of illness in all UzHCs. In UzHCs, each patient received a ticket that served as both a prescription and a record of their diagnosis, medication, and information of diagnostic tests provided by the physician. When patients were diagnosed with any of the selected conditions, the service providers informed our researchers. The researcher systematically selected every second outpatient for participation in the study. For patients under 18 years old or those unable to participate due to illness, an adult caregiver was interviewed instead of the patient. Using this approach, a total of 452 interviews were conducted, with 193 interviews from Hajiganj, 81 from Kachua, 85 from Faridganj, and 93 from Matlab (North) UzHCs. The number of patients interviewed varied across the UzHCs as per sample selection process.

## Study instrument and data collection

We employed semi-structured questionnaires to gather data from health facilities and patients/caregivers. These questionnaires were developed based on existing literature as well as a previous study and underwent review by experts of institutional and ethical review committees [20]. Additionally, the questionnaires were translated into Bengali, piloted in similar environments, and finalized to capture the required information. The questionnaire had several sections covering socioeconomic and demographic information of the respondents and households, details of illness, healthcare utilization and healthcare expenditure for the current illness episode, and patients' opinions on healthcare services provided by the facility.

To estimate the outpatient service costs from the provider's perspective, a health facility inventory survey was conducted with the facility managers, medical staff, accountants, and head assistants. Information was collected through interviews and review of relevant documents, such as the facility register for outpatient service-related statistics, to supplement and complement the survey data, along with financial documents (i.e., budget, expenditure attributable to outpatient) to estimate the total funds received and spent on outpatient services. The inventory survey also collected data on available treatments, costs of medicine, supplies, equipment used for outpatient services, personnel involved, capital items, ingredients used for diagnostic tests, and total number of patients treated.

We conducted a patient exit survey to estimate the user costs for receiving outpatient care for selected most frequent diseases or symptoms. The patient cost information was collected by interviewing the patients or accompanying persons during the exit from the facility. Two data collection teams were deployed to collect data simultaneously across the four selected UzHCs. Each of the team had three trained data collectors and they spent two weeks at each UzHC to interview the outpatients. In this survey, questions were asked on patients' and respondents' demographic characteristics, patients' healthcare utilization, OOPE for treatment at the designated UzHCs and from any other sources within 7 days for the same cause or

diseases, time devoted to care, and household assets. Additionally, a photograph of the advice slip/prescription and drugs/supplies/test report has been taken to extract information on drugs and diagnostic received with the participant's consent. An electronic device-based data collection system called KoboToolbox was used for the face-to-face data collection during February 26, 2021, and March 15, 2021.

## Measuring the cost of treatment

Generally, the total cost of treatment for an illness includes direct, indirect, and intangible costs [21]. Direct costs refer to the financial expenses incurred by health service providers such as costs for human resources, equipment and users costs such as, costs for medicines, diagnostic tests, transportation, food, accommodation, and any informal payments. Indirect costs represent the monetary value of productive time losses to the patient and/or caregivers due to the illness [21]. Intangible costs are associated with the suffering and grief from illness, which are typically challenging to measure due to their subjective nature [21]. However, in this study, we did not consider the intangible costs of illness.

**Direct costs.** Direct costs included both direct medical costs (e.g., registration or ticket fee, medicine, diagnostic tests) and direct non-medical costs (e.g., transportation, food, informal payments, attendant expenditure, and any other types of expenditure relevant to treatment). Since all prescribed medicines were not available at UzHCs' pharmacy, the cost for unavailable medicines were estimated separately as expected medicine expenditure for patients. In this regard, we collected the names and quantities for both the prescribed and provided medicine at UzHCs and estimated the costs using market price. Finally, the OOPE of the different components was estimated and separated into medical and non-medical expenditures for all the interviewed outpatients. Then average of the total medical and non-medical OOPE was estimated per outpatient for different illnesses. From the provider's perspective, we estimated the average treatment costs for outpatient considering the resources utilized to provide the outpatient services. This is because both outpatient and inpatient healthcare services are provided at the UzHCs. Thus, for the provider costs estimation, we included costs of space, furniture, equipment e.g., diagnosis and laboratory test equipment, medicine costs, staff salary, utility costs, institutional costs, and other associated costs required to provide outpatient services. The costs of shared capital items such as staff salary and equipment which are used for treating both inpatient and outpatient care were apportioned considering the proportion used for the outpatients.

**Indirect costs.** We utilized the human capital approach method to estimate the value of potential productivity losses (or income loss as a proxy) as a consequence of illness [18, 22, 23]. The indirect costs for seeking treatment were estimated considering the income loss of patients and their caregivers for both paid and unpaid work (caregiving, household activities). The income loss from foregone non-market activities (unpaid work) was measured using occupation-specific minimum wages from the Bangladesh labor force survey data [24]. Thus, the income loss of patients and caregivers due to treatment -related absences from work were converted into monetary value. In this study, we considered the individual perspective in estimating the indirect cost of treatment.

**Societal costs.** We estimated the disease-specific societal cost of treatment for seeking outpatient healthcare by summing up the total cost from the provider and user perspective.

## Data analysis

We conducted a descriptive analysis and presented the characteristics of the study participants and the most frequent outpatient diseases/symptoms as frequency (n) and percentages (%).

Average costs of treatment from the provider and user perspective were estimated and presented in Bangladeshi Taka (BDT). Average OOPE per patient for outpatient services was estimated in two different scenarios i) actual costs incurred to the patients including payment for diagnostics tests at UzHCs and ii) expected costs for prescribed medicine and diagnostic tests which were not provided from UzHCs. Expected OOPE for medicine was defined as the costs to be incurred by patients if medicines were not provided by the UzHCs and had to be purchased externally. For estimating the medicine costs, the market price (medicines of local pharmaceutical companies) was considered. Likewise, we collected price information from private facilities for the diagnostic tests which were not available at UzHCs. Additionally, we conducted t-tests to understand the statistical difference of average medicine costs estimated as per prescription, cost of medicines provided from UzHCs, and expected costs of medicine to be borne by patients.

We classified the socioeconomic status of the respondents based on their reported possession of durable assets and applied principal component analysis (PCA). The PCA scores were generated using the possession of durable goods (e.g., mobile phones and televisions) [25]. Using the scores generated from PCA, the respondents were divided into five groups. Data were analyzed using Stata/ SE 15.0 and Excel.

## Ethics approval

Ethical approval for the study was obtained from the institutional review board of the International Centre for Diarrhoeal Disease Research, Bangladesh (icddr,b) before data collection (PR#20144). Informed written consent was taken from all the respondents prior to the interviews, and confidentiality and anonymity of their provided information were ensured. All data were collected anonymously and handled confidentially, aligning with the declaration of Helsinki.

## Results

**Table 1** presents the socioeconomic and demographic characteristics of the study participants. About 17% of patients were less than five years old, whereas about 11% were 60 and above. The proportion of female patients (66.8%) was higher compared to the male patients (33.2%). One in four patients had no institutional education whereas 11.3% had higher secondary and above level of education. Among the patients, a sizable percentage was housewives (45.5%). The majority of the patients were married (61.5%) and around 44% of the patient's household had between three to four family members, followed by five to six members (38.7%). About 43% of patients were interviewed from Haziganj UzHC followed by around 21% from Matlab (North) UzHC and the rest of 19% and 18% from Faridganj and Kachua UzHC, respectively.

**Fig 1** illustrates the most common diseases/symptoms at outpatient of the study UzHCs. The commonly reported diseases/symptoms among the patients included the common cold (10.0%), followed by fever (8.0%), and back or chest pain (7.3%). Other notable ailments included respiratory illness (7.1%), gastrointestinal problems (5.1%), and cough (4.4%). The proportion of maternity and gynecological problems such as menstrual problems (5.1%), antenatal care (ANC) (4.2%), urinary tract infection (UTI) (3.8%), and leukorrhea (2.4%) were also noticeable. Chronic conditions such as diabetes were 6.9% of the respondents, while hypertension was 5.5%. Chronic kidney disease was less frequent, accounting for 0.7% of the respondents.

**Table 2** shows the OOPE for outpatient healthcare utilization from the UzHCs. The estimated OOPE per outpatient was BDT 75.9 (95% CI: 66.8–85.0) at UzHCs which included both direct medical and non-medical expenditures. Among the direct medical items, the

**Table 1. Characteristics of outpatient service users.**

| Variables | n = 452 | Percentage (%) |
|---|---|---|
| **Age** | | |
| Less than 5 years old | 77 | 17.0 |
| 5–14 | 42 | 9.3 |
| 15–29 | 108 | 23.9 |
| 30–44 | 102 | 22.6 |
| 45–59 | 74 | 16.4 |
| 60 years and above | 49 | 10.8 |
| **Sex** | | |
| Male | 150 | 33.2 |
| Female | 302 | 66.8 |
| **Education level** | | |
| No institutional education | 113 | 25.0 |
| Up to primary | 131 | 29.0 |
| Secondary | 157 | 34.7 |
| Higher secondary and above | 51 | 11.3 |
| **Occupation** | | |
| Labourer/Worker | 27 | 6.0 |
| Business | 36 | 8.0 |
| Retired/Unemployed | 38 | 8.4 |
| Student | 51 | 11.3 |
| Children | 88 | 19.5 |
| Housewife | 206 | 45.5 |
| Other (maid/self-employed) | 6 | 1.3 |
| **Marital status** | | |
| Married | 278 | 61.5 |
| Unmarried | 144 | 31.9 |
| Widow/ Widower | 27 | 6.0 |
| Divorced/Separated | 3 | 0.6 |
| **Family size** | | |
| Up to 2 members | 30 | 6.7 |
| 3–4 members | 198 | 43.8 |
| 5–6 members | 175 | 38.7 |
| 7 or more members | 49 | 10.8 |
| **Asset quintile** | | |
| Lowest quintile | 91 | 20.1 |
| 2nd quintile | 90 | 19.9 |
| 3rd quintile | 91 | 20.2 |
| 4th quintile | 90 | 19.9 |
| Highest quintile | 90 | 19.9 |
| **Study sites** | | |
| Haziganj UzHC | 193 | 42.7 |
| Kachua UzHC | 81 | 17.9 |
| Faridganj UzHC | 85 | 18.8 |
| Matlab (North) UzHC | 93 | 20.6 |

expenditure for registration and diagnostic tests were BDT 4.7 (95% CI: 4.7–4.8) and BDT 219.1 (95% CI: 93.5–344.6), respectively. Among the non-medical items, the transport expenditure was the major driver (BDT 70.4; 95% CI: 63.9–76.9), followed by expenditure for

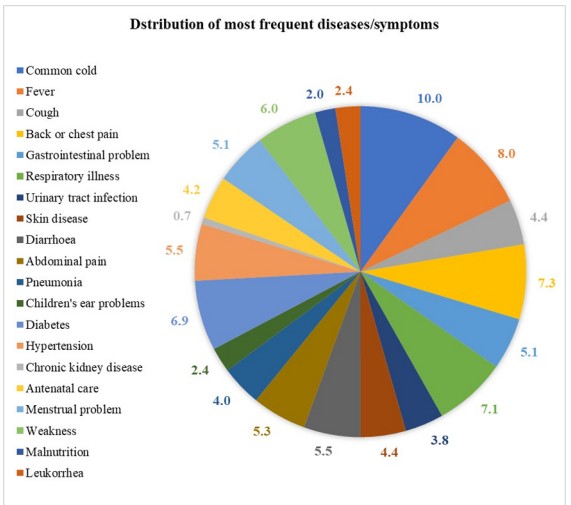

**Fig 1. Distribution of patients by most frequent diseases/symptoms.**

attendants (BDT 56.7: 95% CI: 6.2–107.1) and food (BDT 36.3: 95% CI: 26.3–46.4). While considering the expected OOPE for medicines and diagnostics to be purchased/get from private sources, the expected medicine expenditure per outpatient was BDT 344.0 (95% CI: 307.7–380.3) and diagnostic expenditure was BDT 586.9 (95% CI: 428.6–745.3). The expected diagnostic expenditure was almost three times higher than the average OOPE incurred for

**Table 2. OOPE of respondents for outpatient healthcare utilization at UzHCs including expected medicine and diagnostic expenditure.**

| OOPE components | n (%) | Mean BDT / USD (95% CI) |
|---|---|---|
| **Medical expenditure items** | | |
| Ticket/Registration fee | 274 (60.6) | 4.7 (4.7–4.8) |
| Diagnostic expenditure incurred at UzHCs | 11 (2.4) | 219.1 (93.5–344.6) |
| *Medical expenditure* | **452 (100.0)** | **8.2 (4.1–12.3)** |
| **Non-medical expenditure items** | | |
| Food expenditure | 60 (13.3) | 36.3 (26.3–46.4) |
| Attendant expenditure | 12 (2.7) | 56.7 (6.2–107.1) |
| Transport expenditure | 384 (85.0) | 70.4 (63.9–76.9) |
| Other expenditure (e.g., tips) | 11 (2.4) | 65.0 (13.2–116.8) |
| *Non-medical expenditure* | **452 (100.0)** | **67.7 (60.2–75.2)** |
| **Subtotal** [a] | **452 (100.0)** | **75.9 (66.8–85.0)** |
| **Expected OOPE for medicine and diagnostic** | | |
| Medicine | 385 (98.5) | 344 (307.7–380.3) |
| Diagnostic | 59 (15.1) | 586.9 (428.6–745.3) |
| *Subtotal* | **391 (100.0)** | **427.3 (376.1–478.4)** |
| **Total OOPE** [b] | **452 (100.0)** | **445.5(397.3–493.7)** |
| **Total OOPE in USD** | **452 (100.0)** | **5.2 (4.7–5.8)** |

USD to BDT conversion rate, 1USD = 85.0 BDT

[a] Without OOPE for expected medicine and diagnostic

[b] Including expected OOPE for medicine and diagnostic

diagnostic tests at UzHCs (Prescribed tests that were not available at UzHCs). The expected OOPE for medicine and diagnostics tests was estimated to be BDT 427.3 (95% CI: 376.1–478.4) per outpatient which was more than five times higher than the incurred OOPE at UzHCs. On average, a patient incurred BDT 446 as total OOPE, considering both medical and non-medical expenses, as well as expected costs for medications and diagnostic tests.

Table 3 shows the average costs per outpatient from the provider's perspective. Capital costs per outpatient service were estimated to be BDT 10.5 of which space cost was BDT 9.4. On the other hand, staff salary was identified as a major cost driver for recurrent items with BDT 262.5, followed by others and utilities estimated at BDT 9.8 and BDT 3.8, respectively. However, including both capital and recurrent items, the total estimated cost of the provider was BDT 289.0 for an episode of outpatient service.

Table 4 shows the costs for prescribed medicine, provided medicine, and gap between prescribed and provided medicine across the symptoms/diseases. Expected OOPE for medicine (costs for required medicines to be purchased from outside) was the highest for ANC (BDT 594.6), followed by UTI (BDT 497.2) and for menstrual problems (BDT 474.4). The expected cost was found to be the lowest for children's ear problems at BDT 137.5. The overall expected medicine cost was estimated to be BDT 320.1 per patient. On average, patients bore 73.3% of the total estimated cost of medicines, ranging from 53.6% (for diabetes) to 92.7% (for malnutrition). The expected OOPE for medicine (medicines not provided from the UzHCs) were significantly (p<0.001) higher compared to the estimated costs of prescribed medicine.

Table 5 shows the societal costs of the most common outpatient disease or symptoms in UzHCs. Among the diseases, the societal costs for ANC were the highest (BDT 2019.8; 95% CI: 1554.1–2485.5), followed by costs for diabetes (BDT 1211: 95% CI: 957.9–1464.2) and cough (BDT 1203.9: 95% CI: 915.6–1492.2).

## Discussion

This study aimed to estimate the costs of outpatient healthcare services for common diseases/symptoms in selected UzHCs of Bangladesh. The study highlighted an often-overlooked aspect of OOPE incurred by patients while utilizing PHC at public hospitals. The findings revealed that there is a significant gap between the costs of prescribed and provided medicine at UzHCs ranging between BDT 137 (USD 1.6) to BDT 595 (USD 7.0) per outpatient user for most common diseases/symptoms. This is because, although prescribed medicines at public hospitals

**Table 3. Average costs per outpatient from the provider perspective (n = 4).**

| Cost components | Cost items | Mean BDT / USD |
|---|---|---|
| **Capital items** | Space | 9.4 |
| | Furniture | 0.3 |
| | Equipment | 0.9 |
| **Subtotal of capital** | | **10.5** |
| **Recurrent items** | Staff salary | 262.5 |
| | Cleaning | 2.9 |
| | Utilities | 3.8 |
| | Others | 9.3 |
| **Subtotal of recurrent** | | **278.5** |
| **Total provider cost per patient** | | **289.0** |
| **Total provider cost per patient in USD** | | **3.4** |

USD to BDT conversion rate, 1 USD = 85.0 BDT

**Table 4. Expected cost of medicine and gap between the costs of prescribed and provided medicine in BDT/USD.**

| Diseases/ Symptoms | n (%) | Estimated medicine costs as per prescription (a) | Estimated cost of medicine provided from UzHC (b) | Expected medicine costs to be borne by patients (gap of costs between as per prescribed and provided medicine) (c) | % share of total estimated cost of medicine provided from UzHC | % share of total estimated cost of medicine to be borne by patients |
|---|---|---|---|---|---|---|
| Antenatal care | 19 (5.1) | 727.2 (517.4–937.0) | 132.6 (49.8–215.5) | 594.6 (361.2–827.9) | 18.2 (9.6–23.0) | 81.8 (69.8–88.4) |
| Urinary tract infection | 17 (4.5) | 666.9 (415.3–918.6) | 169.7 (79.9–259.5) | 497.2 (268.0–726.4) | 25.4 (19.2–28.2) | 74.6 (64.5–79.1) |
| Diabetes | 30 (8.0) | 612.9 (443.2–782.6) | 284.3 (188.3–380.3) | 328.6 (192.0–465.2) | 46.4 (42.5–48.6) | 53.6 (43.3–59.4) |
| Pneumonia | 10 (2.7) | 584.5 (400.4–768.6) | 143.4 (-12.1–298.9) | 441.1 (251.3–630.9) | 24.5 (0.0–38.9) | 75.5 (62.8–82.1) |
| Hypertension | 23 (6.1) | 584.0 (349.0–818.9) | 124.0 (63.1–184.9) | 460.0 (237.3–682.6) | 21.2 (18.1–22.6) | 78.8 (68.0–83.4) |
| Respiratory illness | 32 (8.5) | 572.5 (402.1–742.8) | 179.3 (110.0–248.5) | 393.2 (259.5–527.0) | 21.8 (42.8–17.9) | 78.2 (40.1–85.3) |
| Menstrual problem | 21 (5.6) | 535.0 (333.8–736.3) | 60.7 (21.9–99.5) | 474.4 (297.8–651.0) | 11.3 (6.6–13.5) | 88.7 (89.2–88.4) |
| Leukorrhea | 11 (2.9) | 452.7 (286.6–618.9) | 155.1 (41.6–268.5) | 297.6 (123.5–471.8) | 34.3 (14.5–43.4) | 65.7 (43.1–76.2) |
| Back or chest pain | 28 (7.5) | 449.3 (320.4–578.3) | 58.6 (39.0–78.3) | 390.7 (259.7–521.7) | 13.0 (12.2–13.5) | 87.0 (81.1–90.2) |
| Weakness | 27 (7.2) | 387.6 (259.0–516.2) | 82.9 (56.6–109.2) | 304.7 (171.4–438.1) | 21.4 (21.9–21.2) | 78.6 (66.2–84.9) |
| Abdominal pain | 13 (3.5) | 356.6 (214.1–499.2) | 124.9 (9.5–240.3) | 231.7 (108.9–354.4) | 35.0 (4.4–48.1) | 65.0 (50.9–71.0) |
| Skin disease | 19 (5.1) | 354.3 (143.3–565.3) | 73.3 (25.5–121.0) | 281.1 (89.2–472.9) | 20.7 (17.8–21.4) | 79.3 (62.2–83.7) |
| Gastrointestinal problems | 21 (5.6) | 302.5 (197.4–407.6) | 61.8 (40.3–83.3) | 240.7 (132.7–348.7) | 20.4 (20.4–20.4) | 79.6 (67.2–85.5) |
| Cough | 16 (4.3) | 287.3 (177.7–396.9) | 121.6 (56.9–186.2) | 165.8 (92.9–238.6) | 42.3 (32.0–46.9) | 57.7 (52.3–60.1) |
| Common cold | 31 (8.3) | 280.6 (167.8–393.5) | 115.3 (28.9–201.6) | 165.4 (81.8–249.0) | 41.1 (17.2–51.2) | 58.9 (48.7–63.3) |
| Malnutrition | 9 (2.4) | 226.2 (49.4–403.1) | 16.7 (4.5–28.8) | 209.6 (40.7–378.4) | 7.4 (9.1–7.1) | 92.7 (82.4–93.9) |
| Diarrhoea | 14 (3.7) | 223.3 (109.8–336.8) | 81.9 (36.5–127.3) | 141.4 (27.4–255.4) | 36.7 (33.2–37.8) | 63.3 (25.0–75.8) |
| Fever | 26 (6.9) | 219.6 (137.4–301.8) | 54.6 (27.6–81.6) | 165.0 (88.1–241.9) | 24.9 (20.1–27.0) | 75.1 (64.1–80.2) |
| Children's ear problems | 8 (2.1) | 194.9 (127.1–262.7) | 57.4 (3.4–111.4) | 137.5 (68.9–206.1) | 29.5 (2.7–42.4) | 70.5 (54.2–78.5) |
| **Total** | **375 (100.0)** | **436.9 (395.1–478.7)** | **116.9 (99.5–134.2)** | **320.1 (282.6–357.6)** | **26.7 (25.2–28.0)** | **73.3 (71.5–74.7)** |
| **Total (in USD)** | **375 (100.0)** | **5.1 (4.6–5.6)** | **1.4 (1.2–1.6)** | **3.8 (3.3–4.2)** | **0.3 (0.2–0.3)** | **0.9 (0.8–0.9)** |
| **p-value (t-test)** | | ***a vs b p<0.001*** | | ***a vs c p<0.001*** | | |

are meant to be "free", often they are not provided due to unavailability of stock or rules of medicine disbursement. Thus, patients are required to purchase the prescribed medicines and get diagnostics (which are not available at the UzHCs) from private sources.

The average OOPE for outpatient healthcare incurred at the UzHCs was estimated at BDT 75.9 (USD 0.89). On the other hand, when we considered the expected medicine and diagnostic expenditure (as per the prescription and not provided from UzHCs), the overall OOPE increased by five times (BDT 75.9 vs BDT 427.3). We found the costs for diagnostic test was the main driver of the expected OOPE followed by medicine costs. A study conducted by MOHFW showed similar findings on the drivers of OOPE for healthcare in Bangladesh [24]. Moreover, studies in rural Bangladesh [26–28] and other LMICs [29–32] consistently identified medicine and diagnostic costs as the key drivers of OOPE. We found that on average, patients had to bear 73.3% of the total estimated costs of medicine while seeking healthcare from UzHCs. A national-level study estimated that 68.5% of total healthcare expenditure was borne as OOPE by the households in accessing healthcare services of which medicine costs

**Table 5. Societal costs for most frequent outpatient diseases or symptoms in BDT.**

| Most frequent outpatient diseases/symptoms | Patients OOPE | UzHCs Medicine costs | Provider costs (except medicine) | Indirect costs | Societal costs |
|---|---|---|---|---|---|
| | Mean (95% CI) | Mean (95% CI) | Mean (95% CI) | Mean (95% CI) | Mean (95% CI) |
| Antenatal care | 1383.8 (931.8–1835.8) | 132.6 (49.8–215.5) | 286.5 (242.5–330.4) | 216.9 (84.6–349.3) | 2019.8 (1554.1–2485.5) |
| Diabetes | 542.7 (330.6–754.8) | 284.3 (188.3–380.3) | 288.2 (260.6–315.8) | 95.8 (63.6–128.0) | 1211.0 (957.9–1464.2) |
| Urinary tract infection | 671.5 (426.6–916.4) | 169.7 (79.9–259.5) | 260.1 (216.5–303.6) | 102.6 (55.2–149.9) | 1203.9 (915.6–1492.2) |
| Leukorrhea | 604.1 (62.6–1145.6) | 155.1 (41.6–268.5) | 312.2 (270.7–353.6) | 74.1 (52.4–95.8) | 1145.5 (613.8–1677.1) |
| Menstrual problem | 705.1 (461.4–948.7) | 60.7 (21.9–99.5) | 290.8 (262.5–319) | 82.8 (61.2–104.3) | 1139.3 (882.1–1396.6) |
| Hypertension | 568.9 (336–801.7) | 124 (63.1–184.9) | 285.7 (257.8–313.6) | 85.9 (52.3–119.5) | 1064.5 (809.5–1319.4) |
| Respiratory illness | 785.2 (243.9–1326.4) | 179.3 (110–248.5) | 276.9 (255.6–298.3) | 67.7 (41.3–94.1) | 1059.8 (865.4–1254.2) |
| Pneumonia | 493.1 (284.2–702) | 143.4 (-12.1–298.9) | 283.5 (239.8–327.2) | 88.8 (40.2–137.4) | 1008.8 (816.5–1201.1) |
| Back or chest pain | 487.4 (329.8–644.9) | 58.6 (39–78.3) | 281 (251.3–310.8) | 83.1 (60.6–105.7) | 910.2 (754.6–1065.7) |
| Skin disease | 378.8 (157.2–600.5) | 73.3 (25.5–121) | 262.9 (228.1–297.7) | 126.1 (21.4–230.7) | 841.1 (491.2–1190.9) |
| Weakness | 381.1 (247.4–514.9) | 82.9 (56.6–109.2) | 277.8 (251–304.6) | 89.7 (59.0–120.4) | 831.5 (696.3–966.73) |
| Fever | 429.6 (195.9–663.2) | 54.6 (27.6–81.6) | 262.8 (230.6–295) | 65.6 (36.1–95.1) | 812.6 (586.4–1038.8) |
| Gastrointestinal problems | 336.3 (233.2–439.5) | 61.8 (40.3–83.3) | 272 (241.9–302.1) | 113.0 (49.7–176.3) | 783.1 (635.4–930.8) |
| Cough | 281.5 (178.5–384.5) | 121.6 (56.9–186.2) | 249.5 (213.2–285.8) | 120.4 (54.7–186.2) | 772.9 (576.2–969.6) |
| Malnutrition | 284.6 (107.6–461.5) | 16.7 (4.5–28.8) | 300.6 (0.0–0.0) | 163.7 (54.4–273.1) | 765.5 (493.5–1037.6) |
| Abdominal pain | 291.2 (139.6–442.7) | 124.9 (9.5–240.3) | 272.8 (226.8–318.7) | 74.3 (43.5–105.0) | 763.1 (602.2–924.1) |
| Children's ear problems | 227.3 (125.8–328.7) | 57.4 (3.4–111.4) | 323.4 (289.3–357.6) | 88.2 (48.9–127.4) | 696.2 (526.1–866.3) |
| Common cold | 228.8 (141–316.7) | 115.3 (28.9–201.6) | 271.1 (248.2–294.1) | 65.9 (37.4–94.5) | 681.1 (560.3–802.0) |
| Diarrhoea | 205.3 (73.2–337.4) | 81.9 (36.5–127.3) | 264.6 (226.7–302.4) | 90.3 (39.9–140.8) | 642.1 (477.1–807.1) |

had the major share [24]. Such higher OOPE for healthcare is a significant barrier to accessing and utilizing healthcare services in LMICs [33].

We found that the societal costs for ANC (BDT 2020) were the highest among the included diseases followed by diabetes (BDT 1211) and cough (BDT 1204). Several studies have estimated the costs of ANC, which is comparatively lower than our estimates [34–36]. There might be several reasons for this difference in ANC costs. For example, along with the provider costs, we have included the expected OOPE assuming that the patients would pay for the prescribed medicines and diagnostic tests which were not provided from UzHCs. Whereas the other studies either considered only incurred OOPE or did not include infrastructure, capital equipment, overhead or other costs in calculating total costs of the treatment [34].

From the provider's perspective, the average costs per outpatient (excluding medicine costs) was estimated at BDT 289 (USD 3.4), where recurrent costs (96%) was the major costs driver. This finding is similar to a study conducted in Ghana, where the costs per outpatient was estimated at USD 4.51 at district-level hospitals [37].

The lack of provisioning of a full course of free prescribed medicines at UzHCs increases the burden of OOPE for healthcare and raises the tendency of non-compliance among care

seekers, especially among the low-income population. While skipping a few doses might seem harmless in hindsight, it can have serious long-term consequences. Stopping medication prematurely, often when symptoms fade, is a common form of non-compliance and a significant contributor to antimicrobial resistance (AMR) [38]. Studies have documented this link, highlighting non-compliance as a major driver of AMR in Bangladesh [39–42]. Consequently, the inadequate medicine supplies at UzHCs can trigger a series of adverse events, poor quality of life, and ultimately, a heavier burden on healthcare resources due to increased hospital visits and admissions in later period [43–46]. On the other hand, the inadequate mix of human resources and equipment within UzHC's subsidized diagnostic services proved to be insufficient as per the need of the population [47]. This shortfall compelled the patients to seek diagnostic services outside the UzHCs, leading to a rise in OOPE burdens [26].

The study's findings highlighted a critical gap between medicine demand and supply within the PHC system, demanding immediate policy attention. Current policies, which solely rely on inpatient services to allocate resources for UzHCs, neglects the significant patient load at outpatient departments [9]. This inadequate medicine provision as well as facilities for diagnostic services for the common diseases/symptoms further exacerbates the burden of OOPE to the patients. Such high OOPE is a major barrier to accessing healthcare services for the low-income population in Bangladesh. While the government's endeavour to achieve UHC promises universal access to quality healthcare, such gaps significantly hinder healthcare utilization among the low-income population. The study has several limitations to be considered while interpreting the findings. The costs of medicine and diagnostic tests were estimated based on market prices and private providers. Additionally, it was assumed that the patients would get medicines and diagnostics services from private sources that were not provided from UzHCs. Since the costs of medicine and diagnostic services vary at different private sources, the calculated costs may not precisely reflect the exact unit costs. However, the estimates on medicine costs gap will give an idea of how much budget will be required for the provisioning of a full course of medicines for the considered illness. Another significant limitation to consider is related to the study design. Since this is a cross-sectional study and data were collected over a two-month period, seasonal bias may have occurred in the list of common diseases, potentially influencing the average costs estimation per outpatient. Although illness types were selected in consultation with service providers, there may be a presence of response bias. For this reason, The variability in case observations across facilities during the data collection timeframe resulted in certain cases not being observed in all facilities. Despite these limitations, the empirical evidence from this study holds significant strength. The findings are expected to provide valuable insights to policymakers involved in health budget allocation and strengthening the PHC system in Bangladesh. Furthermore, a large-scale, well-planned national prospective study could be undertaken to estimate more precise societal costs of outpatient services at the UzHCs.

## Conclusion

This study estimated the societal costs of outpatient healthcare services for common diseases in selected UzHCs of Bangladesh. Despite the government's obligation to provide free healthcare, a considerable gap exists between prescribed and available medications resulting into high OOPE on patients while seeking outpatient care from UzHCs. Bridging this gap necessitates policy-level changes, including strengthening the existing rural healthcare system by guaranteeing a steady supply of medicines and increasing the number of essential diagnostic tests at the UzHC level. Additionally, curbing the burden of OOPE through optimized resource allocation and expanded health insurance coverage is crucial. A fundamental shift in

the financing of UzHCs requires considering outpatient service demand in the fund allocation mechanism in addition to the number of beds. These measures will help to meet the healthcare needs of the population by improving their access, reducing financial burden, and accelerating Bangladesh's progress towards achieving UHC.

## Supporting information

**S1 File. Socioeconomic and illness data.**
(XLS)

**S2 File. Medicine data.**
(XLS)

**S3 File. Diagnostic data.**
(XLS)

**S4 File. Provider cost data.**
(XLSX)

## Acknowledgments

We sincerely acknowledge the support provided by the Health Economics Unit, Ministry of Health and Family Welfare to conduct this study. We also acknowledge the authority of Haziganj, Kachua, Faridganj, and Matlab North UzHC for their cooperation and support in conducting the survey. icddr,b is thankful to the Governments of Bangladesh and Canada for providing unrestricted/ institutional support.

## Author Contributions

**Conceptualization:** Md. Zahid Hasan, Ziaul Islam, Shehrin Shaila Mahmood.

**Data curation:** Md. Zahid Hasan, Gazi Golam Mehdi, Mohammad Wahid Ahmed.

**Formal analysis:** Md. Zahid Hasan, Gazi Golam Mehdi, Khadija Islam Tisha, Md. Golam Rabbani.

**Funding acquisition:** Ziaul Islam.

**Project administration:** Gazi Golam Mehdi, Mohammad Wahid Ahmed.

**Writing – original draft:** Md. Zahid Hasan, Gazi Golam Mehdi, Khadija Islam Tisha, Md. Golam Rabbani.

**Writing – review & editing:** Md. Zahid Hasan, Gazi Golam Mehdi, Khadija Islam Tisha, Md. Golam Rabbani, Mohammad Wahid Ahmed, Subrata Paul, Ziaul Islam, Shehrin Shaila Mahmood.

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
