## [Decision Letter · Decision Letter 0]

14 Oct 2024

PONE-D-24-19534Costs of outpatient services at selected primary healthcare centers in rural Bangladesh: a cross-sectional studyPLOS ONE

Dear Dr. Hasan,

Thank you for submitting your manuscript to PLOS ONE. After careful consideration, we feel that it has merit but does not fully meet PLOS ONE’s publication criteria as it currently stands. Therefore, we invite you to submit a revised version of the manuscript that addresses the points raised during the review process.

We look forward to receiving your revised manuscript.

Kind regards,

Tope Michael Ipinnimo, MBBS, MPH, FWACP, FMCPH

Academic Editor

PLOS ONE

2. Please provide additional details regarding participant consent. Your study included minors, state whether you obtained consent from parents or guardians. If the need for consent was waived by the ethics committee, please include this information.

“We sincerely acknowledge the funding support provided by the Health Economics Unit, Ministry of Health and Family Welfare. We also acknowledge the authority of Haziganj, Kachua, Faridganj, and Matlab North UzHC for their cooperation and support in conducting the survey. icddr,b is also thankful to the Governments of Bangladesh and Canada for providing unrestricted/ institutional support.”

“The study was funded by the Health Economics Unit (HEU), Health Services Division, Ministry of Health and Family Welfare, the Government of Bangladesh (Grant #: GR-01964).”

5. In the online submission form, you indicated that [The datasets used and/or analyzed during the current study are available from the corresponding author upon reasonable request.].

Additional Editor Comments:

Kindly address the comments and concerns in the attached document

Reviewers' comments:

Reviewer's Responses to Questions

**Comments to the Author**

1. Is the manuscript technically sound, and do the data support the conclusions?

Reviewer #1: Yes

Reviewer #2: Partly

2. Has the statistical analysis been performed appropriately and rigorously? 

Reviewer #1: Yes

Reviewer #2: I Don't Know

3. Have the authors made all data underlying the findings in their manuscript fully available?

Reviewer #1: Yes

Reviewer #2: Yes

4. Is the manuscript presented in an intelligible fashion and written in standard English?

Reviewer #1: Yes

Reviewer #2: No

5. Review Comments to the Author

Reviewer #1: METHODOLOGY

Sample size

Kindly show how you arrived at your sample size of 452

Sampling Technique

Though the authors made some attempts under study site and sampling size/data collection to explain the sampling technique used by this study, the explanation is still not clear. Therefore, kindly create a section titled “Sampling Technique” and explain step by step on how you selected your facilities and respondents. It appears you used a multistage sampling technique and you stated how you purposively selected the primary health centers used, however you did not explain how you allocated questionnaires to each of the centers and the number of questionnaires allocated. You did not also clearly state how and technique used in respondents recruitment over the study period or are these patients also selected using convenience sampling by the physician (a convenience sampling (non-probabilistic technique) may introduce bias into you study and reduce its soundness scientifically).

RESULTS

Table 4

It will be beneficial if an additional column is created to show percentage of cost paid by the patients out of the “Expected Medicine costs as per prescription´ for each diseases and total cost. This will show the percentage OOPE for each diseases and also cumulatively.

DISCUSSION

Include the percentage of OOPE gotten in this study (as gotten in new table 4) in your discussion part from line 297 to 307. This will enable you make clearer comparison with the “national-level study” value quoted on line 306.

CONCLUSION

The conclusion should also reflect your study outcome based on you objectives

Reviewer #2: 1. The present introduction states the purpose of the study and discusses how these findings can add value to the existing evidence. However, it is suggested to highlight a summary of previous research work to indicate gaps.

2. Semi-structured questionnaires were administered, however, no reference was given for information about the number of questions, dimensions, and scoring of the questionnaires. Similarly, there was no reference to the validity and reliability of the research instrument

3. The sampling design and methodology of the research is not quite clear

4. Language structure should be revised to reflect research rigor.

6. PLOS authors have the option to publish the peer review history of their article (what does this mean?). If published, this will include your full peer review and any attached files.

Reviewer #1: **Yes: **TAOFEEK ADEDAYO SANNI

Reviewer #2: No

---

## [Author Response · Author response to Decision Letter 0]

22 Nov 2024

Response to Editor:

Comment 1. Please ensure that your manuscript meets PLOS ONE's style requirements, including those for file naming. The PLOS ONE style templates can be found at

Response: Thanks for sharing the links. We have now formatted the manuscript following the PLOS ONE guideline. 

Comment 2. Please provide additional details regarding participant consent. Your study included minors, state whether you obtained consent from parents or guardians. If the need for consent was waived by the ethics committee, please include this information.

Response: Thank you for the comment. While the study included data for individuals below 18 years of age, minors themselves were not interviewed. Instead, for patients under 18 or those unable to participate due to illness, interviews were conducted with their parents or an adult caregiver (Please see page 7; lines 149-150). We have now revised the ethical consideration sub-section and provided additional details regarding participant consent (Please see page 11; lines 247-252).

Comment 3. We note that the grant information you provided in the ‘Funding Information’ and ‘Financial Disclosure’ sections do not match. When you resubmit, please ensure that you provide the correct grant numbers for the awards you received for your study in the ‘Funding Information’ section.

Response: Many thanks for identifying the inconsistency. We have now checked and made the necessary corrections as suggested. 

Comment 4. Thank you for stating the following in the Acknowledgments Section of your manuscript:

“We sincerely acknowledge the funding support provided by the Health Economics Unit, Ministry of Health and Family Welfare. We also acknowledge the authority of Haziganj, Kachua, Faridganj, and Matlab North UzHC for their cooperation and support in conducting the survey. icddr,b is also thankful to the Governments of Bangladesh and Canada for providing unrestricted/ institutional support.”

“The study was funded by the Health Economics Unit (HEU), Health Services Division, Ministry of Health and Family Welfare, the Government of Bangladesh (Grant #: GR-01964).”

Response: Thanks for the instructions. We have now removed funding-related information from the manuscript and as suggested we have provided the funding information in the cover letter for your information. We appreciate you taking care of the changes to the online submission form on my behalf. (Please see page 21; lines 419-423)

Funding information 

The study was funded by the Health Economics Unit (HEU), Health Services Division, Ministry of Health and Family Welfare, the Government of Bangladesh (Grant #: GR-01964)

Comment 5. In the online submission form, you indicated that [The datasets used and/or analyzed during the current study are available from the corresponding author upon reasonable request.].

Response: Thank you for sharing PLOS journals' data sharing policy. We would like to confirm that the deidentified data used in this manuscript in included as supplementary material. 

Response to Additional Editor:

Response: We have addressed the additional comments under the reviewers’ comments section, as they are similar. 

Responses to Reviewer 1:

Comment on Methodology

Comment 1: Sample size: Kindly show how you arrived at your sample size of 452

Response: Many thanks for this comment. We have arrived at our sample size of 452 based on the following approach “Since the primary focus of this study was on estimating the cost of outpatient service, a single population mean formula was used to determine the sample size. A prior study reported that the standard deviation of cost for outpatient service at public hospitals in Bangladesh was USD 124. We used this value to calculate the sample size for the UzHCs. Assuming a 95% confidence level and a 10% margin of error from the mean costs, we determined that 340 respondents would be required for the exit survey. Further, taking into account a 1.2 design effect and a 10% non-response rate, the estimated sample size increased to 512. This sample was proportionally allocated across the four UzHCs based on the previous year's utilization rates. However, we could interview a total of 452 respondents during the time frame for data collection.” We have also revised the manuscript accordingly (Please see pages 5-6; lines 118-127).

Comment 2: Sampling Technique: Though the authors made some attempts under study site and sampling size/data collection to explain the sampling technique used by this study, the explanation is still not clear. Therefore, kindly create a section titled “Sampling Technique” and explain step by step on how you selected your facilities and respondents. It appears you used a multistage sampling technique and you stated how you purposively selected the primary health centers used, however you did not explain how you allocated questionnaires to each of the centers and the number of questionnaires allocated. You did not also clearly state how and technique used in respondent recruitment over the study period or are these patients also selected using convenience sampling by the physician (a convenience sampling (non-probabilistic technique) may introduce bias into you study and reduce its soundness scientifically).

Response: Thanks for this comment. We have now provided sub-section with a detailed explanation of the sampling technique as follows (Please see pages 6-7; lines 129-153): 

“Sampling technique

We collected cost data through surveys from both service providers and users to capture their perspectives. 

Selection of health facility for survey 

We purposively selected Chandpur district for this study, which has a total of seven UzHCs. These UzHCs were categorized into two groups based on the number of beds: 31-bedded (four) and 50-bedded (three) UzHCs. From each group, we selected two UzHCs based on their outpatient utilization rates—one with high utilization and one with low utilization. The outpatient utilization data for these facilities were extracted from the local health bulletin for the past three years (2018–2020) [19]. The selected 31-bedded UzHCs were Faridganj and Matlab North, while the selected 50-bedded UzHCs were Hajiganj and Kachua. 

Patient selection for exit survey

To select patients for the exit survey, the most frequent outpatient diseases or symptoms were listed by consulting with the Upazila Health and Family Planning Officer and Medical Officers of the selected UzHCs. Each patient received a ticket that served as both a prescription and a record of their diagnosis, medication, and any diagnostic information provided by the physician. When patients presented with any of the selected conditions, the service providers informed our researchers, who systematically selected every second patient utilizing outpatient services for participation in the study. For patients under 18 years old or those unable to participate due to illness, an adult caregiver was interviewed instead. Using this approach, a total of 452 interviews were conducted, with 193 interviews from Hajiganj, 81 from Kachua, 85 from Faridganj, and 93 from Matlab (North) UzHCs. The number of patients interviewed varied across the UzHCs as per sample selection process.”

Comment on Results

Comment 3: Table 4: It will be beneficial if an additional column is created to show percentage of cost paid by the patients out of the “Expected Medicine costs as per prescription´ for each diseases and total cost. This will show the percentage OOPE for each diseases and also cumulatively.

Response: Thank you for your suggestion. We have added two columns: (i) % share of the total estimated cost of medicines provided by UzHC and (ii) % share of the total estimated cost of medicines to be borne by patients. This now shows the percentage of estimated medicine costs covered by UzHC alongside the expected costs to be borne by patients (Please see page 16; Table 4). 

Comment on Discussion

Comment 4: Include the percentage of OOPE gotten in this study (as gotten in new table 4) in your discussion part from line 297 to 307. This will enable you make clearer comparison with

the “national-level study” value quoted on line 306.

Response: Thank you for this suggestion. We have incorporated the percentage of OOPE obtained in this study, as presented in the revised Table 4, into the discussion section (Please see page 18; lines 343-347). This addition allows for a clearer comparison with the national-level study value mentioned on line 345 (previous line 306).

Comment on Conclusion

Comment 5: The conclusion should also reflect your study outcome based on your objectives

Response: Thank you for your feedback. We have revised the conclusion to ensure it reflects the study's objectives and outcomes.

“This study estimated the societal costs of outpatient healthcare services for common diseases in selected UzHCs of rural Bangladesh. Despite the government's obligation to provide free healthcare, a considerable gap exists between prescribed and available medications resulting into high OOPE on patients while seeking outpatient care from UzHCs. A need-based system for the supply of medicine is essential to ensure compliance with the treatment, especially for low-income population. Ensuring a reliable supply of essential medications and expanding diagnostic services at UzHCs will help reduce the financial burden, improve healthcare access and facilitate achieving UHC in Bangladesh.” (Please see page 21; lines 408-415). 

Response to Reviewer 2:

Comment 1. The present introduction states the purpose of the study and discusses how these findings can add value to the existing evidence. However, it is suggested to highlight a summary of previous research work to indicate gaps.

Response: Thank you for your valuable comments and suggestions. We have carefully reviewed the manuscript and emphasized the significance of the study. Additionally, we have cited previous research to highlight existing gaps in the literature (Please see page 5; lines 99-107). However, we acknowledge that there are limited comparable studies in the context of Bangladesh, although we have incorporated all relevant existing studies.

Comment 2. Semi-structured questionnaires were administered; however, no reference was given for information about the number of questions, dimensions, and scoring of the questionnaires. Similarly, there was no reference to the validity and reliability of the research instrument.

Response: Thanks for this comment. We have now provided a clear statement on the study instruments as follows “We employed a semi-structured questionnaire to gather data from health facilities and patients/caregivers. These questionnaires were developed based on existing literature and underwent review by experts, as well as by institutional and ethical review committees. Additionally, the questionnaire was translated into Bengali, contextualized for the Bangladeshi setting, piloted in similar environments, and finalized to ensure that respondents comprehended all the questions………”(Please see pages 7-8; lines 155-183)

Comment 3. The sampling design and methodology of the research is not quite clear. 

Response: Thank you for your comment. We have thoroughly revised the methodology section, including a separate subsection for sample size calculation, sampling techniques, and data collection methods to enhance clarity and detail (Please see pages 5-11; lines 110-254).

Comment 4. Language structure should be revised to reflect research rigor.

Response: Thank you for your suggestion. We have carefully revised the language and structure of the manuscript to enhance its clarity and rigor.

---

## [Decision Letter · Decision Letter 1]

16 Dec 2024

PONE-D-24-19534R1Costs of outpatient services at selected primary healthcare centers in rural Bangladesh: a cross-sectional studyPLOS ONE

Dear Dr. Md. Zahid Hasan,

Thank you for submitting your manuscript to PLOS ONE. After careful consideration, we feel that it has merit but does not fully meet PLOS ONE’s publication criteria as it currently stands. Therefore, we invite you to submit a revised version of the manuscript that addresses the points raised during the review process.

 Kindly download the attached file titled "PONE-D-24-19534" and address the comments and concerns outlined within.

We look forward to receiving your revised manuscript.

Kind regards,

Tope Michael Ipinnimo, MBBS, MPH, FWACP, FMCPH

Academic Editor

PLOS ONE

Additional Editor Comments:

Kindly download the attached file named "PONE-D-24-19534" and attend to the comments and concerns raised within in addition to the reviewers comments below.

Reviewers' comments:

Reviewer's Responses to Questions

**Comments to the Author**

1. If the authors have adequately addressed your comments raised in a previous round of review and you feel that this manuscript is now acceptable for publication, you may indicate that here to bypass the “Comments to the Author” section, enter your conflict of interest statement in the “Confidential to Editor” section, and submit your "Accept" recommendation.

Reviewer #1: All comments have been addressed

Reviewer #2: (No Response)

2. Is the manuscript technically sound, and do the data support the conclusions?

Reviewer #1: Yes

Reviewer #2: Partly

3. Has the statistical analysis been performed appropriately and rigorously? 

Reviewer #1: Yes

Reviewer #2: I Don't Know

4. Have the authors made all data underlying the findings in their manuscript fully available?

Reviewer #1: Yes

Reviewer #2: Yes

5. Is the manuscript presented in an intelligible fashion and written in standard English?

Reviewer #1: Yes

Reviewer #2: Yes

6. Review Comments to the Author

Reviewer #1: A good attempt has been made on all issues raised in the initial review. The corrections made by the author is sufficient enough.

Reviewer #2: Semi-structured questionnaires were administered; however, no reference was given for information about the number of questions, dimensions, and scoring of the questionnaires. Additionally, it is not clear if it is the same questionnaires/research instruments administered to both the facilities and the respondents considering the possible difference in information gathered from the different categories of respondents. Similarly, the validity and reliability of the research instrument is not clearly stated. Without this information, particularly the construct validity, the research is not replicable.

7. PLOS authors have the option to publish the peer review history of their article (what does this mean?). If published, this will include your full peer review and any attached files.

Reviewer #1: **Yes: **TAOFEEK ADEDAYO SANNI

Reviewer #2: No

---

## [Author Response · Author response to Decision Letter 1]

20 Dec 2024

Editorial comments

Many thanks for the comments on the document. However, we have noticed that the comments were provided on the previous version of the manuscript thus most of the comments were addressed in the previous version. 

Comment: Please change to "Introduction"

Response: We have now revised. (Please see page 3, line 55)

Comment: PHC options like Union Health and Family Welfare Centers change like to “such as the”

Response: We have revised the background section. (Please see page 4, lines 75-77)

Comment: Reference for the NHP 2011?

 Response: Yes, the reference is for the National Health Policy 2011. (Please see page 4, line 83)

Comment: What is OOPE? Please define with first use 

Response: OOPE stands for out-of-pocket expenditure. We have abbreviated it in the introduction section, where it was first used. (Please see page 3, line 60) 

Comment: This is not the description of the study site but the sampling technique. Please do a proper description of the surveyed area and move the other information to subheading "sampling technique" Additionally, I'd suggest the authors describe properly how these health facilities were selected. What sampling methods were used?

Response: Many thanks for the suggestions. We have revised the description of settings, sampling technique, and selection of healthcare facilities. (Please see page 5, lines 113-115; page 7, lines 144-154) 

Comment: How did the authors arrive at this sample size? How were these patients selected from the health facilities?

Response: We have updated the sample size calculation in the latest version of the submitted manuscript. Please see the sample size sub-section. (Please see pages 5-6, lines 118-128)

Comment: Why pre-selected symptom/diseases? It would be good to mentioned the criteria used to select these conditions. How were these diseases selected?

Response: Many thanks for the comment. We have now revised the section and added how we prepared the list of diseases. “To select patients for the exit survey, we initially prepared a list of the most frequent outpatient illness or symptoms. The list was prepared in consultation with the Upazila Health and Family Planning Officers and Medical Officers of the selected UzHCs. During the listing

, different facilities reported different illness as most frequent. However, from the collected list, we selected the illness those were reported as common in majority of the UzHCs to keep consistent with the type of illness in all UzHCs.” (Please see page 7, lines 144-149)

Comment: What is the source of this questionnaire? Please provide a reference if it has been published before or attach it as an appendix if it was designed by the authors? Give a more description of the study instrument. Also, give a detail of it validity and reliability.

Response: We have used a standard questionnaire for cost data collection following a previous study conducted by the authors. We have now included a line on the questionnaire and a reference. We have also added the following lines regarding the questionnaire “The questionnaire had several sections covering socioeconomic and demographic information of the respondents and households, details of illness, healthcare utilization and healthcare expenditure for the current illness episode, and patients’ opinions on healthcare services provided by the facility. ” (Please see page 7, lines 164-167)

 Comment “review of relevant documents” This was not mentioned as part of the study design

Response: Many thanks, we have now added a line in the study design section as follows. “The quantitative method included a health facility survey and review of relevant costs document to estimate provider costs and a patient exit survey to assess the costs incurred by patients for utilizing outpatient services” (Please see page 5, lines 114-115)

Comment: Is there any inclusion and exclusion criteria for this study? 

Response: The only inclusion criteria we considered that patients have to be diagnosed with the listed illness and respondents will have to be aged 18 and above which has been included under the sub-section “Patient selection for exit survey”. (Please see page 7, lines 151-156)

Comment: Please remove the "e.t.c." Ditto for other part of the work. List all that was estimated.

Response: Thanks for the comment. We have revised it accordingly. (Please see page 9, lines 194-195)

Comment: Human capital approach or friction cost method Which of them was used?

Response: We have used the human capital approach for costing. We have now revised this accordingly. (Please see page 10, line 222)

Comment: I'd suggest that the USD rate is provided most especially, for the major cost average 

Response: Many thanks for the suggestions. We have included the USD rate for the major cost average.

Comment: Table 1. Occupation (%) These didn't add up to 100. Confirm for the rest of the tables

Response: We have checked and revised the table. (Please see page 13, Table 1)

Comment: Table 1. How was this (asset quintile) analysed? Please include in the methods

Response: Many thanks for the comment. We used principal component analysis to construct the asset quintiles based on the reported durable assets of the respondent. We have now added the following lines in the data analysis section under method. “We classified the socioeconomic status of the respondents based on their reported possession of durable assets and applying principal component analysis (PCA). The PCA scores were generated using the possession of durable goods (e.g., mobile phones and televisions) [25]. Using the scores generated from PCA, the respondents were divided into five groups.” (Please see page 11, lines 255-258)

Comment. Table 4 and 5. It would be easier to follow if this could be arranged in either ascending or descending order

Response: Many thanks for the suggestion. We have now arranged the tables in descending order. (Please see page 17, Table 4; and page 18, Table 5)

Comment. Table 5. CI for Provider costs (except medicine) in the malnutrition row? 

Response: One limitation of our study is the variability in case observation across facilities during data collection. Although illness types were selected in consultation with service providers, certain cases were not observed in all facilities. For instance, the "Malnutrition" category, highlighted in your comment, was recorded in only one facility (Haziganj UzHC). Consequently, the CI for provider costs in this category could not be calculated. We have also highlighted this as one of the limitations of the study. (Please see page 22, lines 416-418)

Comment: “a sizable number of patients (about 85%) still needed to purchase medicine and diagnostic services.” Is this part of the study findings? Kindly reference it if not.

Response: We have revised and removed this statement in the updated version.

Comment: “It was estimated that the diagnostic services cost is the main driver of the expected OOPE followed by medicine costs which are justified by a study conducted by MoHFW [23].” Additionally, this finding could be further compared with studies done in other parts of the world including sub-Saharan African such as 

1. https://dx.doi.org/10.4314/jcmphc.v34i1.3, 

2. https://www.ajol.info/index.php/jcmphc/article/view/139376, 

3. doi: 10.4103/ijcm.ijcm_431_22 and 

4. doi: 10.18311/jhsr/2018/21335

Response: We have compared this finding with studies conducted in other LMICs, including Ethiopia, the Democratic Republic of Congo, and Nigeria and included the suggested references. (Please see page 19, lines 350-352)

Comment: “Bridging this gap necessitates policy-level changes, including strengthening the existing rural healthcare system by guaranteeing a steady supply of medicines and increasing the number of essential diagnostic tests at UzHCs level.” From here till the end of the paragraph are recommendations. Please move it and combine with the one at the conclusion section.

Response: Many thanks for the suggestion. We have revised it accordingly. (Please see page 22, lines 419-420)

Comment: “A fundamental shift towards a demand-driven allocation model in the healthcare financing strategy is required.” The allocation of resources seems to be in-patients service demand driven already. I would suggest "taking into considering out-patient service demand in resource allocation" instead.

Response: Thanks for the suggestion, we have now revised the statement accordingly. However, the current resource allocation in UzHC is primarily based on number of facility beds and staffing levels, which do not adequately reflect the resources required as per the need of facility utilization by inpatients or outpatients. (Please see page 22, lines 419-420)

Comment: Conclusion: “...having potential side effects including medication non-adherence, antimicrobial resistance, and poor health outcomes.” These are not study findings. Kindly delete.

Response: We have now revised and deleted this from the conclusion. (Please see page 22, lines 415-422)

Comment: Conclusion: “…a demand-driven”. Please revise as suggested above.

Response: Many thanks for the suggestion. We have revised it accordingly. (Please see page 22, lines 419-420)

Reviewer comments

Reviewer #2

Comment: Semi-structured questionnaires were administered; however, no reference was given for information about the number of questions, dimensions, and scoring of the questionnaires. Additionally, it is not clear if it is the same questionnaires/research instruments administered to both the facilities and the respondents considering the possible difference in information gathered from the different categories of respondents. Similarly, the validity and reliability of the research instrument is not clearly stated. Without this information, particularly the construct validity, the research is not replicable.

Response: We have used a standard questionnaire for cost data collection following a previous study conducted by the authors. We have now included a line on the questionnaire and a reference. (Reference: Ahmed, S., de Broucker, G., Hasan, M. Z., Mehdi, G. G., Martin del Campo, J., Constenla, D., Patenaude, B., & Uddin, M. J. A.-B. & M. G. F. (2020). Cost of measles in children under 5 in Bangladesh (2017-18) (V2 ed.). Harvard Dataverse. https://doi.org/doi:10.7910/DVN/ZXZEUY90) (Please see page 7, lines 164-167). We used separate questionnaires to collect data from health facilities and patients/caregivers. The details are elaborated in the "Study Instrument and Data Collection" subsection 

(Please see pages 7-8, lines 159-187)

---

## [Editor Report · Decision Letter 2]

26 Dec 2024

Costs of outpatient services at selected primary healthcare centers in rural Bangladesh: a cross-sectional study

PONE-D-24-19534R2

Dear Dr. Md. Zahid Hasan,

We’re pleased to inform you that your manuscript has been judged scientifically suitable for publication and will be formally accepted for publication once it meets all outstanding technical requirements.

Kind regards,

Tope Michael Ipinnimo, MBBS, MPH, FWACP, FMCPH

Academic Editor

PLOS ONE

---

## [Editor Report · Acceptance letter]

3 Jan 2025

PONE-D-24-19534R2 

PLOS ONE

Dear Dr. Hasan, 

I'm pleased to inform you that your manuscript has been deemed suitable for publication in PLOS ONE. Congratulations! Your manuscript is now being handed over to our production team.

Kind regards, 

on behalf of

Dr. Tope Michael Ipinnimo 

Academic Editor

PLOS ONE